# *Alnus sibirica* Compounds Exhibiting Anti-Proliferative, Apoptosis-Inducing, and *GSTP1* Demethylating Effects on Prostate Cancer Cells

**DOI:** 10.3390/molecules26133830

**Published:** 2021-06-23

**Authors:** Seo-Yeon Seonu, Min-Ji Kim, Jun Yin, Min-Won Lee

**Affiliations:** Laboratory of Pharmacognosy and Natural Product Derived Medicine, College of Pharmacy, Chung-Ang University, Seoul 06974, Korea; tjdus4593@gmail.com (S.-Y.S.); kam4256@naver.com (M.-J.K.); yinjun89@naver.com (J.Y.)

**Keywords:** *Alnus sibirica*, hirsutenone, prostate cancer, cancer prevention, DNA methylation

## Abstract

*Alnus sibirica* (AS) is distributed in Korea, Japan, China, and Russia and has reported anti-oxidant, anti-inflammatory, and reducing activities on atopic dermatitis-like skin lesions, along with other beneficial health properties. In the present study, we tried to prove the cancer-preventive activity against prostate cancer. The extracted and isolated compounds, oregonin (**1**), hirsutenone (**2**), and hirsutanonol (**3**), which were isolated from AS, were tested for anti-proliferative activity. To do this, we used the MTT assay; NF-κB inhibitory activity, using Western blotting; apoptosis-inducing activity using flow cytometry; DNA methylation activity, using methylation-specific polymerase chain reaction in androgen-dependent (LNCaP) and androgen-independent (PC-3) prostate cancer cell lines. The compounds (**1**–**3**) showed potent anti-proliferative activity against both prostate cancer cell lines. Hirsutenone (**2**) exhibited the strongest NF-κB inhibitory and apoptosis-inducing activities compared with oregonin (**1**) and hirsutanonol (**3**). DNA methylation activity, which was assessed for hirsutenone (**2**), revealed a concentration-dependent enhancement of the unmethylated DNA content and a reduction in the methylated DNA content in both PC-3 and LNCaP cells. Overall, these findings suggest that hirsutenone (**2**), when isolated from AS, may be a potential agent for preventing the development or progression of prostate cancer.

## 1. Introduction

Prostate cancer (PCa) is one of the most common cancers in men and typically develops after the of age 65 years [1]. The high and increasing incidence rates of PCa across the world [2] have motivated research to find new agents that can effectively prevent PCa. There are various risk factors involved in PCa, including family history, hormones, race, ageing, diet, obesity, inflammation, and several environmental factors [2,3]. Additionally, prostate cancer is not a disease that should only concern the male population. It is important that men of all ages pay attention to PCa, and attempt to manage it in advance from a young age. Prostate cancer in young men represents a distinct clinical phenotype and can be predicted for early reference [4].

Diverse natural products and phytochemicals have been reported to interfere with the effects of targeting such factors, and can have direct PCa prevention effects [5,6,7,8,9]. *Alnus sibirica* Fisch. ex Turcz. (AS) is distributed in Korea, Japan, Russia, and China and has been widely used for traditional medicine [10] owing to its various bioactivities, including anti-oxidant, anti-inflammatory, melanogenesis-inhibitory [11], reducing activities on atopic dermatitis-like skin lesions [12,13] and anti-adipogenic [14] activities. In addition, various phytochemicals have been isolated from *Alnus* species, such as tannins [15,16,17], diarylheptanoids [18,19], flavonoids [20], and triterpenoids.

We screened 213 different kinds of local native plants for the ability to inhibit LNCaP, an androgen-sensitive prostate cancer cell line, and DU145, which is a more aggressive androgen-insensitive prostate cancer cell line [21]. We isolated several diarylheptanoids, which are characteristic components of the *Alnus* species [22,23,24]. Additionally, we reported cytotoxic activities of diarylheptanoids from the *Alnus* species on various human and mouse cancer cell lines. [25]. Furthermore, we found that hirsutenone inhibits the activation of NF-κB involved in human mammary epithelial cells stimulated with TPA [26]. In this study, we evaluated the anti-proliferative activities and inhibitory activities on NF-κB, apoptosis and DNA methylation of three diarylheptanoids (**1**–**3**), isolated from AS on androgen-dependent (LNCaP) and androgen-independent (PC-3) prostate cancer cell lines. This was conducted through a series of experiments in order to determine their potential value of preventing PCa, which may contribute to the chemo-preventive effects exerted by these phytochemicals.

## 2. Results

### 2.1. Anti-Proliferative Activity

Oregonin (**1**), hirsutenone (**2**) and hirsutanonol (**3**) were isolated from AS [22] (Figure 1) and we examined the anti-proliferative activity using two PCa cell lines (PC-3, LNCaP). Among the three compounds, MTT assay demonstrated that hirsutenone (**2**) had the most potent anti-proliferative activity (Table 1).

### 2.2. NF-κB Inhibitory Activity

We evaluated the NF-κB inhibitory activity of the three diarylheptanoids, isolated from AS, on PC-3 and LNCaP cells. Western blotting demonstrated that the relative protein expression level of NF-κB was effectively reduced by treatment of both PCa cell lines with oregonin (**1**), hirsutenone (**2**) and hirsutanonol (**3**) (Figure 2).

### 2.3. Apoptosis-Inducing Activity

We evaluated the apoptosis-inducing activity of the three compounds (**1**–**3**) at various concentrations against the two PCa cell lines using flow cytometry. The oregonin (**1**) and hirsutenone (**2**) showed more potent apoptosis-inducing activities than hirsutanonol (**3**) (Figure 3).

### 2.4. DNA Methylation Activity

We examined DNA methylation activity in PCa cells using methylation-specific PCR. Hirsutenone (**2**) showed an enhancement of the unmethylated DNA content and a potent reduction in the methylated DNA content in both PC-3 and LNCaP cells (Figure 4).

## 3. Discussion

Observance of the 2018 World Cancer Research Fund/American Institute for Cancer Research (WCRC/AICR) cancer prevention recommendations and their relationship to PCa were evaluated. A total of 398 PA cases and 302 controls were included in the study [27].

In this study, we tested the PCa prevention effects including anti-proliferative activity, NF-κB inhibitory activity, apoptosis-inducing activity and DNA demethylation activity of the three diarylheptanoids isolated from AS in vitro.

The anti-proliferative activity of these three compounds were evaluated using two PCa cell lines, PC-3 and LNCaP. Among the three compounds, MTT assay demonstrated that hirsutenone (**2**) showed the most potent anti-proliferative activity against both PC-3 and LNCaP cells [IC50, 65.53 on PC-3 and 109.14 on LNCaP] (Table 1).

Nuclear factor kappa-light-chain enhancer of activated B cells (NF-κB) is a well-known negative regulatory factor of apoptosis, which controls DNA transcription, cytokine production, and cell survival. Dysregulation of NF-κB has been associated with inflammation, immune disease, and cancer [28,29,30,31,32]. In particular, the activation of NF-κB signaling correlates with PCa progression in PCa cell lines and has been associated with more advanced stages, chemoresistance, and metastatic spread [33,34,35,36].

Our previous research reported that hisutenone inhibits lipopolysaccharide-activated NF-κB-induced inflammatory mediator production. We investigated the effect of hirsutenone on lipopolysaccharide-induced inflammatory mediator production in keratinocytes in relation to the Toll-like receptor 4-mediated activation of the extracellular signal-regulated kinase (ERK) and NF-κB pathways [37].

The NF-κB inhibitory activity of the three diarylheptanoids (**1**–**3**) in PC-3 and LNCaP cells from Western blotting demonstrated that the relative protein expression level of NF-κB was more effectively reduced by the treatment of both PCa cell lines with oregonin (**1**) and hirsutenone (**2**), than by treatment with hirsutanonol (**3**) (Figure 2).

Apoptosis is a programmed form of cell death that differs from necrosis, which is a form of traumatic cell death induced by external factors, including disease [38,39]. Unlike necrosis, which may result in the uncontrolled release of inflammatory factors, apoptosis does not usually lead to inflammation [40]. Apoptosis plays a crucial role in tumor suppression and is considered to be the most useful target for cancer therapy [41,42].

The apoptosis-inducing activity at various concentrations towards the two PCa cell lines used flow cytometry; **1**–**3** showed apoptosis-inducing activity. The lower left part represents a viable cell, the lower right part indicate early apoptosis. The upper right part is late apoptosis, which indicates cells that have died due to apoptosis progressing and necrosis. The upper left part is a cell necrosis or pathologically dead part. **1**–**3** all showed apoptotic activities. Especially when set at the concentration of 50 μM on PC-3, oregonin (**1**) and hirsutenone (**2**) have more potent apoptosis-inducing activities than hirsutanonol (**3**) (Figure 3).

DNA methylation is a chemical process that results in the addition of methyl groups to a DNA molecule. In general, DNA methylation occurs only on the cytosines of CpG dinucleotides. In healthy somatic cells, approximately 80% of the CpG dinucleotides are methylated and excluded from the defined regions [43,44,45]. However, there are some CpG-rich sequences in somatic cells, called CpG islands, that are usually demethylated [46,47]. In addition, methylated cytosine may convert to thymine by spontaneous deamination, resulting in a T/G permanent mismatch mutation due to the complementary pairing of T with A. When DNA methylation occurs in promoter regions that are located near a gene transcription start site containing a CpG island, gene expression is inhibited, resulting in gene silencing [48,49,50]. Accumulating evidence has pointed to a potentially more important role of gene silencing than that of mutation in cancer. Indeed, several genes have been identified to be silenced or activated in various cancer types, and a large proportion of gene silencing results from DNA methylation. Some methylated genes, such as GSTP1, APC, RASSF1, CD44, and CDH1, have been associated with PCa [51]. In particular, methylation of GSTP1, which encodes an enzyme that is considered to play a key role in cancer susceptibility [52], has been detected in various cancers but shows high specificity for PCa and has been suggested to serve as a clinical biomarker for PCa diagnosis [53,54,55,56,57,58,59].

DNA methylation activity in PCa cells using methylation-specific PCR, hirsutenone (**2**) was selected to evaluate its effect because **2** showed strongest anti-proliferative, NF-κB inhibitory, and apoptosis-inducing activities. DNA methylation activity of hirsutenone (**2**), demonstrated its potent concentration-dependent enhancement of the unmethylated DNA content and reduction in the methylated DNA content in both PC-3 and LNCaP cells. (Figure 4).

Conclusively, we demonstrated that hirsutenone (**2**), one of the active ingredients among compounds of AS, is the most effective for prevention of cancer. These results suggested that **2** is a promising agent to treat and prevent PCa. However, an in vivo experiment will be required.

## 4. Materials and Methods

### 4.1. Plant Material

*Alnus sibirica* Fisch. Ex Turcz. (AS) leaves were collected from Mt. Cheonggye in Gwacheon, Gyeonggi province, Korea, in June 1998. Its identity was confirmed by Prof. M.W. Lee (Pharmacognosy Lab, Laboratory of Pharmacognosy and Natural Product Derived Medicine, College of Pharmacy, Chung-Ang University) [22].

### 4.2. General Experimental Procedure

Precoated silica gel 60 F_254_ plate (Merck, Darmstadt, Germany) was used as thin layer chromatography (TLC). The spots were detected by 10% H_2_SO_4_ by heating and spraying.

The column chromatography was conducted using Amberlite XAD-2 (20–50 mesh, Fluka), Sephadex LH-20 (75–230 μm, mesh, Pharmacia, Uppsala, Sweden), MCI-gel CHP 20P (75–150 μm, Mitsubishi, Tokyo, Japan), ODS-A gel (230/70, 400/230, 500/400 mesh, YMCco, Kyoto, Japan).

IR spectrometer was used Shimazu IR-435 (Kyoto, Japan) and NMR spectrometer (Varian GEMINI 2000 (Phoenix, AZ, USA), 300 MHz), EI-Mass spectrometer (GC-MS/MS-DS, TSQ 700 (Waltham, MA, United States), Negative FAB-Mass spectrometer (VG70-VSEQ (Manchester, UK), Polarimeter (Jasco DIP-370 (Tokyo, Japan)).

### 4.3. Phytochemicals

The air-dried stem barks of AS (5.6 kg) were extracted with 80% Me_2_CO at room temperature, five times. Concentration was performed by removing the Me_2_CO under vacuum. After Me_2_CO evaporation, it was dissolved in H_2_O and filtered. The filtrate was enforced to an Amberlite XAD-2 with a H_2_O:MeOH (from 100:0 to 0:100), yielding 3 subfractions. In fraction of 2, using a Sephadex LH-20 column with a solvent gradient system of H_2_O:MeOH (from 100:0 to 0:100), yielded 3 subfractions (2-1, 2-2, 2-3). Fraction 2-2 was applied to column chromatography MCI gel CHP-20P column, and yielded compound (**1**) (3 g) and compound (**2**) (68 mg). Fraction 3 was subjected to YMC ODS gel column with gradient system of H_2_O: MeOH (from 100:0 to 0:100), yielded compound (**3**) (60 mg).

Compound **1** ${\left[\alpha \right]}_{D}^{20}$: −17.5° (c = 1.0, Me_2_CO), IR $\nu_{max}^{KBr}$cm^−1^: 3367 (OH), 1701 (C=O), 1605, 1522 (Aromatic C=C), Negative FAB MS *m*/*z*: 447 [M − H]^−^, ^1^H-NMR (300 MHz, acetone-d_6_ + D_2_O): δ 6.71–6.76 (4H in total, H-2′,2′′,5′,5′′), 6.54 (2H in total, m, H-6′,6′′), 4.33 (1H, d, *J* = 7.8 Hz, xyl-1), 4.16 (1H, m, H-5), 3.87 (1H, m, xyl-5e), 3.54 (1H, m, xyl-4), 3.39 (1H, t, *J* = 9.0 Hz, xyl-3), 3.21 (1H, m, xyl-5a), 3.13 (1H, m, xyl-2), 2.52–2.86 (8H in total, H-1,2,4,7), 1.77–1.82 (2H in total, m, H-6), ^13^C-NMR (75 MHz, acetone-d_6_ + D_2_O): δ 210.2 (C-3), 146.1 (C-3′, C-3′′), 144.5 (C-4′′), 144.3 (C-4′), 135.4 (C-1′′), 134.4 (C-1′), 120.9 (C-6′), 120.8 (C-6′′), 116.8 (C-5′′), 116.7 (C-5′), 116.5 (C-2′′), 116.4 (C-2′), 104.4 (xyl-1), 78.0 (xyl-3), 76.4 (C-5), 75.1 (xyl-2), 71.3 (xyl-4), 66.8 (xyl-5), 48.4 (C-4), 46.4 (C-2), 38.6 (C-6), 31.6 (C-7), 29.4 (C-1).

Compound **2** ${\left[\alpha \right]}_{D}^{20}$: −25.2° (c = 1.0, Me_2_CO), IR $\nu_{max}^{KBr}$cm^−1^: 3365 (OH), 1650 (-CH=CH-CO-), 1606, 1519 (Aromatic C=C), EI MS: *m*/*z* 328 [M]^+^, ^1^H-NMR (300 MHz, acetone-d_6_): δ 6.93 (1H, dt, *J* = 16.0, 8.0, H-5), 6.73–6.77 (4H in total, m, H-2′,2′′,5′,5′′), 6.51–6.56 (2H in total, m, H-6′,6′′), 6.13 (1H,d, *J* = 16.0 Hz, H-4), 2.47–2.82 (8H in total, m, H-1,2,6,7), ^13^C-NMR (75 MHz, acetone-d_6_): δ 201.5 (C-3), 148.5 (C-5), 146.2 (C-3′, C-3′′), 144.4 (C-4′, C-4′′), 134.2 (C-1′′), 133.9 (C-1′), 131.7 (C-4), 120.7 (C-6′, C-6′′), 116.7 (C-5′, C-5′′), 116.5 (C-2′, C-2′′), 42.6 (C-2), 35.4 (C-6), 34.6 (C-7), 29.5 (C-1).

Compound **3** ${\left[\alpha \right]}_{D}^{20}$: −12.4° (c = 1.0, Me_2_CO), IR $\nu_{max}^{KBr}$cm^−1^: 3336 (OH), 1699 (C=O), 1605, 1519 (Aromatic C=C), EI MS: *m*/*z* 346 [M]^+^, ^1^H-NMR (300 MHz, acetone-d_6_): δ 6.71–6.75 (4H in total, m, H-2′,2′′,5′,5′′), 6.51–6.54 (2H in total, m, H-6′,6′′), 4.06 (1H, m, H-5), 2.52–2.76 (8H in total, m, H-1,2,4,7), 1.68–1.70 (2H in total, m, H-6), ^13^C-NMR (75 MHz, acetone-d_6_): δ 210.7 (C-3), 145.1 (C-3′′), 145.0 (C-3′), 143.3 (C-4′′), 143.1 (C-4′), 134.0 (C-1′′), 133.0 (C-1′), 119.6 (C-6′′), 119.5 (C-6′), 115.7 (C-5′), 115.5 (C-5′′), 115.4 (C-2′′), 115.3 (C-2′), 67.0 (C-5), 50.0 (C-4), 45.1 (C-2), 39.3 (C-6), 31.0 (C-7), 28.7 (C-1).

### 4.4. Cell Culture

The PCa cell lines PC-3 and LNCaP clone FGC (LNCaP.FGC) were purchased from the Korean Cell Line Bank (KCLB, Seoul, Korea). The PC-3 passage number was 28 and LNCaP was 24. These cells were grown at 37 °C in a humidified atmosphere (5% CO_2_) in Roswell Park Memo-rial Institute (RPMI) 1640 medium (Sigma-Aldrich, St. Louis, MO, USA) supplemented with 10% fetal bovine serum, 100 IU/mL penicillin G (Gibco BRL, Grand Island, NY, USA). The cells were used after being counted using a hemocytometer.

### 4.5. Anti-Proliferative Activity

The PC-3 or LNCaP cells (approximately 10^5^ cells/well) were seeded in 96-well plate, pre-incubated for 24 h and then treated with 20 μL of the samples (100, 50, 10, 5, 1 μM of each sample) with serum free DMEM and incubated for a further 24 h at 37 °C. Next, the culture medium was replaced with 100 μL phosphate-buffered saline (PBS) containing 1 mg/mL of 3-(4,5-dimethylthiazol-2-yl)-2,5-diphenyltetrazolium-bromide (MTT), and the cells were incubated for a further 4 h. After the supernatant was removed, the MTT-formazan product was dissolved in 200 μL dimethyl sulfoxide, and the absorbance of the solution at 540 nm was measured using a microplate reader (TECAN, Salzburg, Austria) to determine the degree of cell proliferation.

### 4.6. Flow Cytometry Analysis of Apoptosis

PC-3 and LNCaP cells (approximately 10^6^ cells/well) were seeded in a 6-well plate and pre-incubated for 24 h, and then treated with 200 μL of the samples (500, 250, 125 μM of each sample) with serum free DMEM and incubated for a further 24 h at 37 °C, and the cells were washed once in PBS. Next, the cells were resuspended in binding buffer (100 μL) and stained with fluorescein isothiocyanate-annexin V (5 μL) and propidium iodide (5 μL) (BD, Franklin Lakes, NJ, USA) for 15 min in the dark with vortex mixing. Fluorescence was analyzed using a fluorescence-activated cell sorter (BD), and the percentages of necrotic cells, early and late apoptotic cells, and viable cells were compared.

### 4.7. Protein Extraction and Western Blotting

PC-3 and LNCaP cells (approximately 10^6^ cells/well in 6-well plate) were pre-incubated for 24 h and then treated with 200 μL of the samples (500, 250, 125 μM of each sample) with serum free DMEM and incubated for a further 24 h at 37 °C. The cells were harvested and washed twice with PBS and lysed using 45 μL RIPA buffer (Thermo Fisher Scientific, Waltham, MA, USA) and 5 μL protease inhibitor cocktail (Thermo Fisher Scientific). The cell lysates were kept on ice for 5 min and centrifuged at 15,000 rpm for 15 min at 4 °C. The concentrations of the extracted proteins in the supernatants were determined using a BCA Protein Assay Kit (Thermo Fisher Scientific). Equal concentrations of extracted proteins were resolved on 10% SDS-polyacrylamide gel and the separated proteins were transferred onto nitrocellulose membranes (Bio-Rad Laboratories, Hercules, CA, USA), which were incubated in blocking buffer (Thermo Fisher Scientific) for 1 h to prevent non-specific binding. Subsequently, the membranes were incubated with primary antibodies (1:1000, Bio-Rad Laboratories and Sigma-Aldrich, St. Louis, MO, USA) for 2 h at room temperature (25 °C), washed with Tris-buffered saline-Tween buffer, and incubated with horseradish peroxidase-conjugated secondary antibodies (1:1000, Santa Cruz Biotechnology, Santa Cruz, CA, USA) for 1 h at room temperature. After three washings, an enhanced chemiluminescence detection kit (Santa Cruz Biotechnology) was applied to expose the membranes to X-rays and obtain protein band images on a medical X-ray film (AGFA, Mortsel, Belgium). Band densities were analyzed using the ImageJ software (National Institutes of Health, Bethesda, MD, USA) to quantify relative protein expression.

### 4.8. DNA Extraction and Bisulfite Conversion

PC-3 and LNCaP cells (approximately 10^6^ cells/well) were seeded in 6-well plates and incubated for 24 h and then treated with 200 μL of the samples (1000, 500, 250 μM of each sample) with serum free DMEM and incubated for a further 24 h at 37 °C. After washing with PBS, the cells were centrifuged for 5 min, and DNA was extracted using the QIAamp DNA mini kit (QIAGEN, Hilden, Germany). The concentrations of extracted DNA were measured using a Nanodrop spectrophotometer, and 5 μg of DNA was subjected to bisulfite conversion using the Genetic MethylEasyXceed Rapid DNA Bisulphite Modification Kit (Genetic Signatures, New South Wales, Australia), according to the manufacturer’s instructions.

### 4.9. Methylation-Specific Polymerase Chain Reaction (PCR)

The bisulfited DNA was amplified using the following sets of unmethylated and methylated reaction primers as previously reporte [53]: unmethylated forward primer, 5′-AAAGAGGGAAAGGTTTTTTTGGTTAGTTGTGTGGTG-3′ and reverse primer, 5′-AAACTCCAACAAAAACCTCACAACCTCCA-3′; methylated forward primer, 5′-GGTTTTTTTCGGTTAGTTGCGCGGCG-3′ and reverse primer, 5′-CCAACGAAAACCTCGCGACCTCCG-3′.

The bisulfited DNA was reverse-transcribed to complementary DNA (cDNA) at 70 °C for 1 h in a reaction mixture containing reverse transcriptase, RT buffer, 10 mM dNTP (dNTP mix), oligo dT primer, and RNase inhibitor. The obtained cDNA sample was then amplified using a 2× ReddyMix PCR Master Mix, 10 pM each of upstream and downstream primers, diethyl pyrocarbonate distilled water, and Universal SYBR Green Supermix in a real-time PCR thermocycler (Bio-RAD). The primers used for PCR were chemically synthesized using a DNA synthesizer (Bioneer, Daejeon, Korea) under the following amplification conditions: denaturation at 95 °C for 5 min and second cycles of denaturation at 95 °C for 30 s, annealing at 59 °C for 30 s and extension at 72 °C for 30 s. The final extension was performed at 72 °C for 10 min.

### 4.10. Statistical Analysis

All data are presented as the mean ± standard deviation (SD). Data were compared between the two groups using Student’s *t* test, and among more than two groups using one-way analysis of variance (Student–Newman–Keuls test); *p* < 0.05 was considered to indicate a statistically significant difference. All statistical analyses were performed using the IBM SPSS Statistics program (IBM Corp., Armonk, NY, USA).

## Figures and Tables

**Figure 1 molecules-26-03830-f001:**
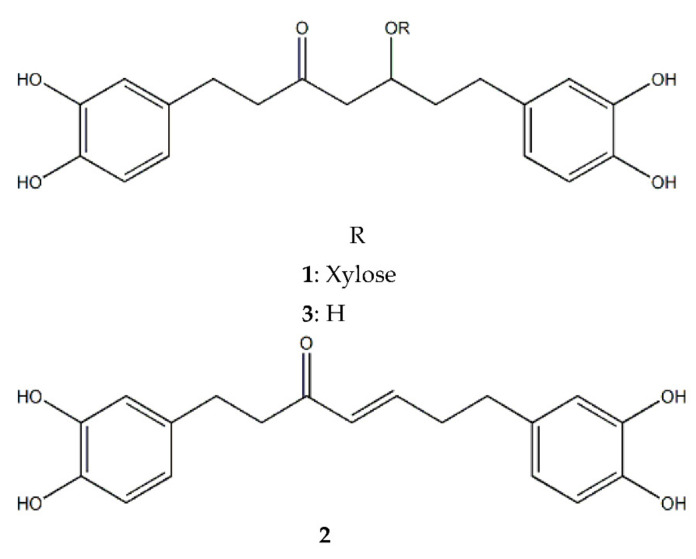
Structures of the three diarylheptanoids isolated from *Alnus sibirica*, oregonin (**1**); hirsutenone (**2**); hirsutanonol (**3**).

**Figure 2 molecules-26-03830-f002:**
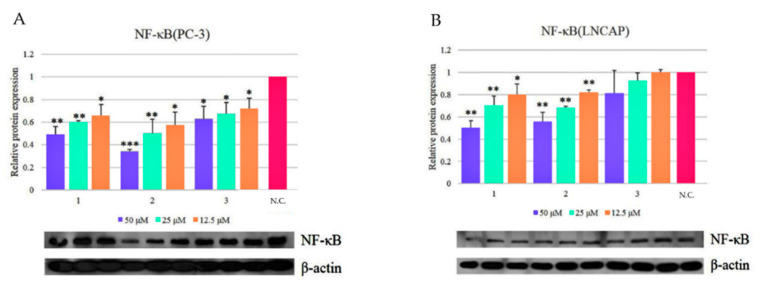
NF-kB inhibitory activities of oregonin (**1**), hirsutenone (**2**), and hirsutanonol (**3**) in PCa cell lines. A: PC-3 cell; B: LNCaP cell. The results were expressed as mean ± SD of triplicate experiments (*n* = 3). NC: Normal control group. *: *p* < 0.05, **: *p* < 0.01, ***: *p* < 0.001, compared with the normal control group.

**Figure 3 molecules-26-03830-f003:**
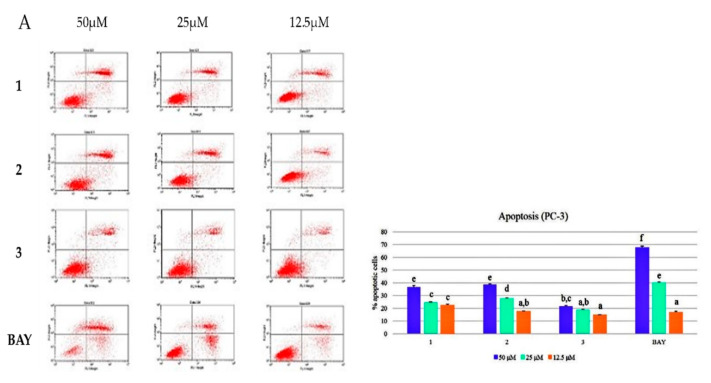
Apoptosis-inducing activities of various concentrations of oregonin (**1**), hirsutenone (**2**), and hirsutanonol (**3**) in PCa cell line. (**A**): PC-3 cells; (**B**): LNCaP cell. The results are expressed as mean ± SD of triplicate experiments (*n* = 3). **BAY**: positive control. Letters a–o in the same column indicate significant differences, *p* < 0.05.

**Figure 4 molecules-26-03830-f004:**
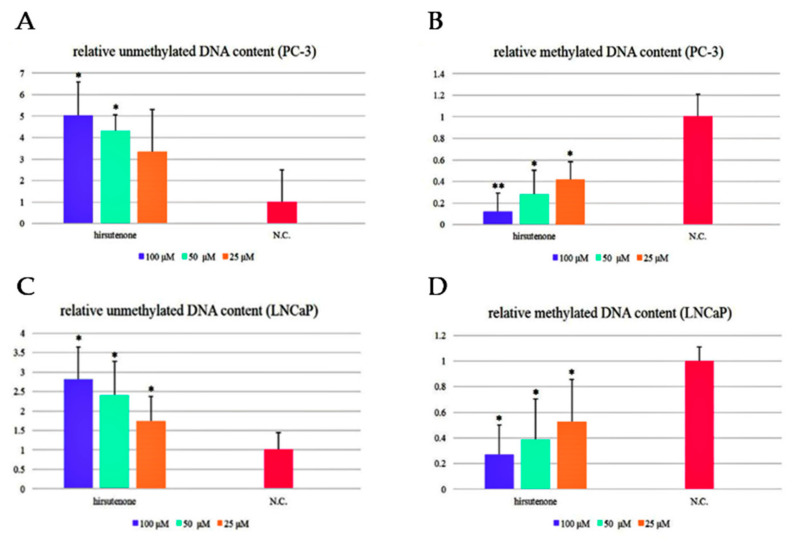
DNA methylation activity of hirsutenone (**2**) in PCa cell lines. (**A**,**C**): relative unmethylated DNA contents in PC-3 and LNCaP cell lines; (**B**,**D**): relative methylated DNA contents in PC-3 and LNCaP cell lines. The results were expressed as the mean ± SD of triplicate experiments (*n* = 3). NC: Normal control group. *: *p* < 0.05, **: *p* < 0.01, compared with the normal control group.

**Table 1 molecules-26-03830-t001:** Anti-proliferative activity of the three diarylheptanoids (**1**–**3**) isolated from *Alnus sibirica* against PC-3 and LNCaP cells (IC_50_^a^, μM).

Compound	PC-3	LNCaP
Oregonin (**1**)	160.46 ± 11.20	199.91 ± 105.08
Hirsutenone (**2**)	65.52 ± 1.71	107.95 ± 18.08
Hirsutanonol (**3**)	129.59 ± 43.16	163.55 ± 25.25

^a^IC_50_: half-maximal inhibitory concentration.

## Data Availability

The data presented in this study are available on request from the corresponding author.

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
