# Peer review of "Alnus sibirica* Compounds Exhibiting Anti-Proliferative, Apoptosis-Inducing, and *GSTP1* Demethylating Effects on Prostate Cancer Cells"

_molecules, 2021, doi:10.3390/molecules26133830_

Round 1

Reviewer 1 Report

The paper “Alnus sibirica compounds have anti-proliferative, apoptosis-inducing, and GSTP1 demethylating effects on prostate cancer cells“ is in the scope of the journal Molecules. In focus of the paper are three diarylheptanoides isolated from Alnus sibirica, plant whose extracts have long been used in Korea, Japan, China and Russia to treat various condition. In this paper autors tried to prove cancer preventive activity of three isolated compounds against prostate cancer. Authors use standard nomenclature and the SI system of units are used consequently. Abstract state the major findings and conclusions of the paper. I suggest this paper to be published in Molecules after suitable revision.

Below I cite few minor (specific) comments:

Results

  • ln 51-54 transform sentence
  • Figure 3. include representative dot-plot diagrams of negative control
  • ln 126 instead Tab.1 it should be Table 1
  • ln 158 instead ...showed potent apoptosis-inducing... it should be ...showed more potent apoptosis-inducing...

References

  • see the Reference List and Citations Guide and cite references as described (e.g. Abbreviated Journal Name....)

Author Response

■ Figure 3. include representative dot-plot diagrams of negative control

→ We have already put Negative control group in NF-kB inhibitory activity in Figure.2 and we demonstrated that compounds (1-3) have inhibitory effects at PC-3 and LNCaP cell and thought that the pattern of concentration-dependent compounds would be sufficient with positive group.

■ ln 126 instead Tab.1 it should be Table 1

→ As you suggested the abbreviation ‘Tab.’ all changed to ‘Table.’

■ ln 158 instead ...showed potent apoptosis-inducing... it should be ...showed more potent apoptosis-inducing...

→ We changed the paragraph by your suggestion. So we have revised this part as below to make it clear.

“The apoptosis-inducing activity at various concentrations towards the two PCa cell lines using flow cytometry, 1-3 showed apoptosis-inducing activity. Especially at the concentration 50μM on PC-3, oregonin (1) and hirsutenone (2) have more potent apoptosis-inducing activities than hirsutanonol (3). (Fig. 3)”

■ see the Reference List and Citations Guide and cite references as described (e.g. Abbreviated Journal Name....)

→ We re-arranged the references according to the form through the reference program ‘endnote’ recommended by Molecules.

Reviewer 2 Report

The article presents the results of an original research on an extract of Alnus sibirica and three pure compounds isolated from this plant: oregonin, hirsutenone, and hirsutanonol. These natural products were tested for their anti-proliferative, NF-κB inhibitory, and pro-apoptotic activities, as well as for their ability to reduce DNA methylation in prostate cancer cell lines.

The introduction provides a brief background on the topic but should be updated with additional, more recent references.

Materials and methods: The entire article makes no mention of the plant part used for the extraction, nor of the nature and polarity of the solvent, except for the remark in line 196:

“The plant materials and compound isolation procedures have been described previously”citing reference: Jeong, D. W., Kim, J. S., Cho, S. M., Lee, Y. A., Kim, K. H., Kim, S. W. and Lee, M. W., 2000. Diarylheptanoids from the stem bark of Alnus hirsuta var. sibirica. Kor. J. Pharmacogn. 31(1): 28-33 (2000).

Without going into great details, some notions of the plant part and extraction are needed in order to make this article independently readable.

Investigation of anti-proliferative, NF-κB inhibitory, pro-apoptotic activities and ability to reduce DNA methylation correspond to standard experiments in the field.

Results are presented clearly and adequately illustrated.

Discussions should avoid general definitions of well-known phenomena like the difference between apoptosis and necrosis; references should be updated.

English language should be checked by a native speaker. Expressions like “anti-atopic dermatitis activities”, “ DNA methylation activity” and “ anti-allergic contact dermatitis” must be correctly rephrased.

Author Response

■ English language should be checked by a native speaker. Expressions like “anti-atopic dermatitis activities”, “ DNA methylation activity” and “ anti-allergic contact dermatitis” must be correctly rephrased.

→ We will have English editing from MDPI before publishing.

→ And we changed reducing activities on atopic dermatitis-like skin lesions instead of anti-atopic dermatitis activities and anti-allergic contact dermatitis with revised reference [12].

→ The terminology of DNA methylation activity is used generally in scientific edition [42, 45, 50, 51, 54]

■ The introduction provides a brief background on the topic but should be updated with additional, more recent references.

→ We changed the paragraph by your suggestion. So we have revised this part as below.

“And prostate cancer is not a disease that only old men should care about, but the prostate cancer that men of all ages need to pay attention to and should be cared for and managed in advance from their youth. Prostate cancer in young men represents a distinct clinical phenotype and is predictable for early reference.”

■ Materials and methods: The entire article makes no mention of the plant part used for the extraction, nor of the nature and polarity of the solvent, except for the remark in line 196:

“The plant materials and compound isolation procedures have been described previously”citing reference: Jeong, D. W., Kim, J. S., Cho, S. M., Lee, Y. A., Kim, K. H., Kim, S. W. and Lee, M. W., 2000. Diarylheptanoids from the stem bark of Alnus hirsuta var. sibirica. Kor. J. Pharmacogn. 31(1): 28-33 (2000). Without going into great details, some notions of the plant part and extraction are needed in order to make this article independently readable.

→ We added in detail the Alnus sibirica material and the method of extraction and isolation used in this experiment and we further describe its compounds sources as below.

“4.1 Plant material and Phytochemical

Alnus sibirica Fisch. ex Turcz. (AS) leaves were collected from Mt. Cheonggyesan in Gwacheon, Gyeonggi province, Korea in June 1998 and its identity was confirmed by Prof. M.W. Lee (Pharmacognosy Lab, Laboratory of Pharmacognosy and Natural Product De-rived Medicine, College of Pharmacy, Chung-Ang University). And the plant materials and compound isolation procedures have been described previously [20].

4.2. Phytochemicals

4-2-1. Compound 1 - [α] 20D: -17.5° (c=1.0, Me2OH), IR ν KBr max cm-1: 3367 (OH), 1701 (C=O), 1605, 1522 (Aromatic C=C), Negative FAB-MS m/z: 447 [M-H]-, 1H-NMR (300 MHz, Me2OH-d6+D2O) : 6.71-6.76 (4H in total, H-2′,2′’,5′,5′′), 6.54 (2H in total, m, H-6′,6′′), 4.33 (1H, d, J =7.8 Hz, xyl-1), 4.16 (1H, m, H-5), 3.87 (1H, m, xyl-5e), 3.54 (1H, m, xyl-4), 3.39 (1H, t, J=9.0 Hz, xyl-3), 3.21 (1H, m, xyl-4), 3.39 (1H, t, J=9.0 Hz, xyl-3), 3.21 (1H, m, xyl-5a), 3.31 (1H, m, xyl-2), 2.52-2.86 (8H in total, H-1,2,4,7), 1.77-1.82 (2H in total, m, H-6), 13C-NMR (75 MHz, Me2OH-d6+D2O)

4-2-2. Compound 2 - Brown amorphous powder, [α] 20D: -25.2° (c=1.0, Me2OH), IR ν KBr max cm-1: 3365 (OH), 1650 (-CH=CH-CO-), 1606, 1519 (Aromatic C=C), EI MS : m/z 328 [M]+, 1H-NMR (300 MHz, Me2OH-d6) : δ 6.93 (1H, dt, J=16.0, 8.0 H-5), 6.73-6.77 (4H in total, m, H-2′,2′′,5′,5′′), 6.51-6.56 (2H in total, m, H-6′,6′′), 6.13 (1H,d, J=16.0 Hz, H-4), 2.47-2.82 (8H in total, m, H-1,2,6,7), 13C-NMR (75 MHz, Me2OH-d6)

4-2-3. Compound 3 - Brown amorphous powder, [α] 20D: -12.4° (c=1.0, Me2OH), IR ν KBr max cm-1: 3336 (OH), 1699 (C=O), 1605, 1519 (Aromatic C=C), EI MS : m/z 346 [M]+, 1H-NMR (300 MHz, Me2OH-d6) : 6.71-6.75 (4H in total, m, H-2′,2′′,5′,5′′), 6.51-6.54 (2H in total, m, H-6′,6′′), 4.06 (1H, m, H-5), 2.52-2.76 (8H, in total, m, H-1,2,4,7), 1.68-1.70 (2H, in total, m, H-6), 13C-NMR (75 MHz, Me2OH-d6)”

Reviewer 3 Report

The article is interesting, although some aspects should be improved.
Authors should emphasise the novelty of their study.
It seems that the authors conducted a lot of experiments, and the descriptions are very laconic and short, both in terms of methodology and results.
According to Molecules Instructions for Authors:
https://www.mdpi.com/journal/molecules/instructions
“In the text, reference numbers should be placed in square brackets [ ], and placed before the punctuation; for example [1], [1–3] or [1,3]”,
while the authors cite the names and the year in the text. Also Reference list is not according to Molecules demands.
In the main title the word ‘have’ sounds wrong.
There are many technical errors in the text.

Introduction:
Give research hypotheses.

Materials and Methods
4.2. Cell culture
Give more details to Cell cultures (detaching, viability testing, passaging etc.). Give passage number of cell lines. Give their full names.
4.3. MTT
There is no description of how the compounds were tested. In what concentrations, how long, how they were seeded ...
Why 105 of cells were seeded in each well 96-well plate? Justify in the text.
How was the IC50 value determined?
Paragraphs 4.4. – 4.7.
The paragraphs should briefly describe the purpose of these experiments to be logical and continuous.
Paragraphs 4.4.-4.7.
There is no description of how the compounds were tested. In what concentrations, how long, how they were seeded ...
Why such concentrations were selected to these studies? On the basis of what. Give justification in the text.

Results
Figure 1 – give references in the caption.
Table 1: give statistics. What does it mean “> 200”? Give full names of tested compounds.
Table is not according to Molecules demands.
Figure 4: The charts are crookedly fit with the rest.
The figures are out of focus.

Conclusions
In what doses the tested compounds are cytotoxic to tested cancer cells?
The authors should include changes in the text on all comments from the reviewer.

Author Response

■ Authors should emphasise the novelty of their study.

→ In discussion, we emphasized the anti-proliferative activity using two PCa cell lines, PC-3 and LNCaP. And we also evaluated the NF-κB inhibitory activity of the three diarylheptanoid (1-3) in PC-3 and LNCaP cells from Western blotting demonstrated that the relative protein expression level of NF-κB and the apoptosis-inducing activity at various concentrations towards the two PCa cell lines using flow cytometry of three compounds (1-3) from Alnus sibirica. Along with DNA methylation activity in PCa cells using methylation-specific PCR.

Finally, we can conclude hirsutenone (2), one of the active ingredients among compounds of AS, is the most effective for prevention of cancer.

■ It seems that the authors conducted a lot of experiments, and the descriptions are very laconic and short, both in terms of methodology and results.

→ We added in detail the Alnus sibirica material and the method of extraction and isolation used in this experiment and we further describe its compounds sources as below.

“4.1 Plant material and Phytochemical

Alnus sibirica Fisch. ex Turcz. (AS) leaves were collected from Mt. Cheonggyesan in Gwacheon, Gyeonggi province, Korea in June 1998 and its identity was confirmed by Prof. M.W. Lee (Pharmacognosy Lab, Laboratory of Pharmacognosy and Natural Product De-rived Medicine, College of Pharmacy, Chung-Ang University) [20].

4.2. Extraction and Isolation

The air-dried stem barks of A. sibirica (5.6 kg) were extracted with 80% Me2Co at room temperature, 5 times. Concentration from removing the Me2Co under vacuum. After Me2Co evaporation, it was dissolved in H2O and filtered. The filtrate was enforced to an Amberlite XAD-2 (20-50 μm, 10 kg, 70 × 50 cm). Chromatography and eluted with a solvent gradient system H2O : MeOH (from 100:0, 50:50 to 0:100 volume ratio), yielding 3 subfractions. In fraction of 2, using a Sephadex LH-20 column (25-100 μm, 2000 g, 10 × 120 cm) with a solvent gradient system of H2O : MeOH (from 100:0 to 0:100 volume ratio), yielded 3 subfractions (2-1, 2-2, 2-3). Fraction 2-2 was applied to column chromatography MCI gel CHP-20P column (50 μm, 150 g, 3 × 50 cm), and yielded compound (1) (3 g) and compound (2) (68mg). Fraction 3 was subjected to YMC ODS gel column (50 μm, 80 g, 2 × 30 cm) with gradient system of H2O : MeOH (from 100:0 to 0:100), yielded compound (3) (60 mg).

4-2-1. Compound 1 - Brown amorphous powder, [α] 20D: -17.5° (c=1.0, Me2OH), IR ν KBr max cm-1: 3367 (OH), 1701 (C=O), 1605, 1522 (Aromatic C=C), Negative FAB-MS m/z: 447 [M-H]-, 1H-NMR (300 MHz, Me2OH-d6+D2O) : 6.71-6.76 (4H in total, H-2′,2′’,5′,5′′), 6.54 (2H in total, m, H-6′,6′′), 4.33 (1H, d, J =7.8 Hz, xyl-1), 4.16 (1H, m, H-5), 3.87 (1H, m, xyl-5e), 3.54 (1H, m, xyl-4), 3.39 (1H, t, J=9.0 Hz, xyl-3), 3.21 (1H, m, xyl-4), 3.39 (1H, t, J=9.0 Hz, xyl-3), 3.21 (1H, m, xyl-5a), 3.31 (1H, m, xyl-2), 2.52-2.86 (8H in total, H-1,2,4,7), 1.77-1.82 (2H in total, m, H-6), 13C-NMR (75 MHz, Me2OH-d6+D2O)

4-2-2. Compound 2 - Brown amorphous powder, [α] 20D: -25.2° (c=1.0, Me2OH), IR ν KBr max cm-1: 3365 (OH), 1650 (-CH=CH-CO-), 1606, 1519 (Aromatic C=C), EI MS : m/z 328 [M]+, 1H-NMR (300 MHz, Me2OH-d6) : δ 6.93 (1H, dt, J=16.0, 8.0 H-5), 6.73-6.77 (4H in total, m, H-2′,2′′,5′,5′′), 6.51-6.56 (2H in total, m, H-6′,6′′), 6.13 (1H,d, J=16.0 Hz, H-4), 2.47-2.82 (8H in total, m, H-1,2,6,7), 13C-NMR (75 MHz, Me2OH-d6)

4-2-3. Compound 3 - Brown amorphous powder, [α] 20D: -12.4° (c=1.0, Me2OH), IR ν KBr max cm-1: 3336 (OH), 1699 (C=O), 1605, 1519 (Aromatic C=C), EI MS : m/z 346 [M]+, 1H-NMR (300 MHz, Me2OH-d6) : 6.71-6.75 (4H in total, m, H-2′,2′′,5′,5′′), 6.51-6.54 (2H in total, m, H-6′,6′′), 4.06 (1H, m, H-5), 2.52-2.76 (8H, in total, m, H-1,2,4,7), 1.68-1.70 (2H, in total, m, H-6), 13C-NMR (75 MHz, Me2OH-d6)”

→ We further revised the result part as below.

“We evaluated the apoptosis-inducing activity of the three AS compounds (1-3) at vari-ous concentrations towards the two PCa cell lines using flow cytometry. The lower left part, represents viable cell, the lower right part indicate early apopto-sis. The upper right part is late apoptosis, which indicates cells that have died due to apoptosis pro-gressing and necrosis, which represents the upper left part, is a cell necrosis or patho-logically dead part. 1-3 showed late apoptotic activities. And the oregonin (1) and hir-sutenone (2) showed more potent apoptosis than hirsutanonol (3). (Fig. 3)”

■ According to Molecules Instructions for Authors:

https://www.mdpi.com/journal/molecules/instructions

“In the text, reference numbers should be placed in square brackets [ ], and placed before the punctuation; for example [1], [1–3] or [1,3]”, while the authors cite the names and the year in the text. Also Reference list is not according to Molecules demands.

Also Reference list is not according to Molecules demands.

→ We re-arranged the references according to the form through the reference program ‘endnote’ recommended by Molecules.

■ In the main title the word ‘have’ sounds wrong.

→ We changed the title as below.

Alnus sibirica compounds exhibiting anti-proliferative, apoptosis-inducing, and GSTP1 demethylating effects on prostate cancer cells”

■ 4.2. Cell culture

Give more details to Cell cultures (detaching, viability testing, passaging etc.). Give passage number of cell lines. Give their full names.

→ We have added details about the cell and cultures information of PC-3 and LNCaP used in the experiment, and cell information shows that this cell is based on prostate cancer as below.

“The PCa cell lines PC-3 and LNCaP clone FGC; LNCaP.FGC were purchased from the Ko-rean Cell Line Bank(KCLB). The origin of PC-3 cell is prostate; grade 4; metastasis to bone and the species is human-male, 62 years old, Caucasian. The KCLB used RPMI1640 with L-glutamine (300mg/L), 25mM HEPES and 25mM NaHCO3, 90%; heat inactivated fetal bovine serum (FBS), 10% media . The origin of LNCaP clone FGC; LNCaP.FGC cell is prostate and the species is human-male, 50 years old, Caucasian. The KCLM used RPMI1640 with L-glutamine (300mg/L), 25mM HEPES and 25mM NaHCO3, 90%; heat inactivated fetal bovine serum (FBS), 10% media. After the passaging 28, PC-3 and pas-saging 47, LNCaP cells were incubated at 37°C in an atmosphere containing 5% CO2 in Roswell Park Memorial Institute (RPMI) 1640 medium (Sigma-Aldrich, St. Louis, MO, USA) supplemented with 10% fetal bovine serum and 100 IU/mL penicillin G (Gibco BRL, Grand Island, NY, USA) for cell culture. And the passage number of LNCaP.FGC used in the experiment is 24.”

■ 4.3. MTT: Why 105 of cells were seeded in each well 96-well plate? Justify in the text.

How was the IC50 value determined?

■ Paragraphs 4.4. – 4.7.

The paragraphs should briefly describe the purpose of these experiments to be logical and continuous.

■ There is no description of how the compounds were tested. In what concentrations, how long, how they were seeded ...

Why such concentrations were selected to these studies? On the basis of what. Give justification in the text.

→ About concentration and cells decision, our laboratory usually dilute the concentration from 100μM and seeding 105 of cells first before conducting the experiment. Afterwards, if the conditions are determined to be insufficient, we are changed the conditions or number of cells. But when it is reasonable, the results are keep determine. So we thought the concentration 100 μM, 50 μM, 25 μM and seeding 105 cells were considered appropriate when conducting the experiment and we revised the paragraph as below.

“4.4 Anti-proliferative activity

Before the biological assay, the cytotoxicity was measured by the mitochondrial-dependent reduction of 3-(4,5-dimethylthiazol-2-yl)-2,5-diphenyltetrazolium-bromide (MTT) (Sigma, St. Louis, MO, USA) to formazan. The PC-3 or LNCaP cells (approximately 105 cells/well) were seeded in 96-well plate and incubated for 24 h in an atmosphere containing 5% CO2 at 37°C and the number of cells are considered sufficient. And the cells were treated with 20 μL of the samples with serum free DMEM and incubated 24 h at 37 °C. Next, the culture medium was replaced with 100 μL phosphate-buffered saline (PBS) containing 1 mg/mL of 3-(4,5-dimethylthiazol-2-yl)-2,5-diphenyl tetrazolium bromide (MTT), and the cells were incubated for a further 4 h and the conditions are considered sufficient. After the supernatant was removed, the MTT-formazan product was dissolved in 200 μL dimethyl sulfoxide, and the absorbance of the solution at 540 nm was measured using a microplate reader (TECAN, Salzburg, Austria) to determine the degree of cell proliferation.”

→ We used GraphPad Prism program for the calculation of IC 50.

→ We added the purpose of experiments (4.4-4.7) as below.

“4.4 Anti-proliferative activity

The anti-proliferative activity of 1-3 from Alnus sibirica using two PCa cell lines, PC-3 and LNCaP were evaluated to know the susceptibility on the cancer cells. Before the biological assay, the cytotoxicity was measured by the mitochondrial-dependent reduction of 3-(4,5-dimethylthiazol-2-yl)-2,5-diphenyltetrazolium-bromide (MTT) (Sigma, St. Louis, MO, USA) to formazan. The PC-3 or LNCaP cells (approximately 105 cells/well) were seeded in 96-well plate and incubated for 24 h in an atmosphere containing 5% CO2 at 37°C and the number of cells are considered sufficient. And the cells were treated with 20 μL of the samples with serum free DMEM and incubated 24 h at 37 °C. Next, the culture medium was replaced with 100 μL phosphate-buffered saline (PBS) containing 1 mg/mL of 3-(4,5-dimethylthiazol-2-yl)-2,5-diphenyl tetrazolium bromide (MTT), and the cells were incubated for a further 4 h and the conditions are considered sufficient. After the supernatant was removed, the MTT-formazan product was dissolved in 200 μL di-methyl sulfoxide, and the absorbance of the solution at 540 nm was measured using a microplate reader (TECAN, Salzburg, Austria) to determine the degree of cell proliferation.

4.5 Flow cytometry analysis of apoptosis

The apoptosis-inducing activity of 1-3 using two PCa cell lines, PC-3 and LNCaP from flow cytometry were evaluated to know the possibility of cancer prevention. PC-3 and LNCaP cells were seeded in a 6-well plate and incubated for 24 h, as described above. The cells were then harvested and washed once in PBS. Next, the cells were resuspended in binding buffer (100 μL) and stained with fluorescein isothiocyanate-annexin V (5 μL) and propidium iodide (5 μL) (BD, Franklin Lanes, NJ, USA) for 15 min in the dark with vortex mixing. Fluorescence was analyzed using a fluorescence-activated cell sorter (BD), and the percentages of necrotic cells, early and late apoptotic cells, and viable cells were compared.

4.6 Protein extraction and western blotting

The NF-κB inhibitory activity of 1-3 in PC-3 and LNCaP cells from Western blotting were evaluated to know the relation of appoptosis. After PC-3 and LNCaP cells were incubated for 24 h, the cells were washed twice with ice-cold PBS and lysed using 45 μL RIPA buffer (Thermo Fisher Scientific, Waltham, MA, USA) and 5 μL protease inhibitor cocktail (Thermo Fisher Scientific). The cell lysates were kept on ice for 5 min, vortexed once every minute, and centrifuged at 15,000 rpm for 15 min at 4°C. The concentrations of the extracted proteins in the su-pernatants were determined using a BCA Protein Assay Kit (Thermo Fisher Scientific). Equal concentrations of extracted proteins were resolved on 10% sodium dodecyl sulfate (SDS)-polyacrylamide gel electrophoresis gels, comprising distilled water, 1.5 M Tris-HCl, 30% acrylamide/bis-acrylamide, 10% SDS, 10% APS, and TEMED (Bio-Rad Laboratories, Hercules, CA, USA). The separated proteins were transferred onto ni-trocellulose membranes (Bio-Rad Laboratories), which were incubated in blocking buffer (Thermo Fisher Scientific) for 1 h to prevent non-specific binding. Subsequently, the membranes were incubated with primary antibodies (1:1000, Bio-Rad Laboratories and Sigma-Aldrich, St. Louis, MO, USA) for 2 h at room temperature (25°C), washed with Tris-buffered saline-Tween buffer, and incubated with horseradish peroxi-dase-conjugated secondary antibodies (1:1000, Santa Cruz Biotechnology, Santa Cruz, CA, USA) for 1 h at room temperature. After three washings, an enhanced chemilu-minescence detection kit (Santa Cruz Biotechnology) was applied to expose the mem-branes to X-rays and obtain protein band images on a medical X-ray film (AGFA, Mortsel, Belgium). Band densities were analyzed using the ImageJ software (National Institutes of Health, Bethesda, MD, USA) to quantify relative protein expression.

4.7. DNA extraction and bisulfite conversion

DNA methylation activity in PCa cells using methylation-specific PCR were evaluated to confirm the activities as cancer chemoprevention. PC-3 and LNCaP cells were seeded in 6-well plates and incubated for 24 h. After washing with PBS, the cells were centrifuged for 5 min, and DNA was ex-tracted using the QIAamp DNA mini kit (QIAGEN, Hilden, Germany). The concentra-tions of extracted DNA were measured using a Nanodrop spectrophotometer, and 5 μg of DNA was subjected to bisulfite conversion using the Genetic MethylEasyXceed Rapid DNA Bisulphite Modification Kit (Genetic Signatures, NSW, Australia), according to the manufacturer’s instructions.”

Round 2

Reviewer 2 Report

The authors updated their paper according to the suggestions of the reviewer. As the article presents the bioactivity of natural products which are less common in the scientific literature, it  should represent an interesting addition to the journal’s collection of publications.

Author Response

■ The authors updated their paper according to the suggestions of the reviewer. As the article presents the bioactivity of natural products which are less common in the scientific literature, it should represent an interesting addition to the journal’s collection of publications. → We added hypotheses about the AS with prostate cancer in Introduction as below. “We isolated several diarylheptanoids, which are characteristic components of the Alnus species [20]. And we screened 213 different kinds of local native plants for the ability to inhibit LNCaP, an androgen-sensitive prostate cancer cell line, and DU145, which is a more aggressive androgen-insensitive prostate cancer cell line [21]. And we reported cytotoxic activities of diarylheptanoids from Alnus species on various human and mouse cancer cell lines. [22]. And we also found that hirsutenone inhibits the activation of NF-κB involved in human mammary epithelial cells stimulated with TPA [23]. In this study, we evaluated inhibitory activities of proliferation, NF-κB, apoptosis and DNA methylation of three diarylheptanoids (1-3) isolated from AS on androgen-dependent (LNCaP) and androgen-independent (PC-3) prostate cancer cell lines through a series of experiments to determine their potential value of preventing PCa which may contribute to the chemopreventive effects exerted by these phytochemicals.”

Reviewer 3 Report

The authors did not respond to all comments of the reviewer.

Still the descriptions especially in the methodology and results is very chaotic.

There are many technical errors in the text (double spaces, lack of spaces etc.).

English language must be corrected by a native speaker.

Introduction:

Give research hypotheses – still not added.

Materials and Methods

4.3. Cell culture

The description is very chaotic and disordered.

If both cell lines are grown similarly, there is no need to describe the culturing method separately, just mention the differences.

From the description now it is not clear how many cell lines were tested?

4.4. MTT

The description is very chaotic and disordered.

Still there is no description of how the compounds were tested. In what concentrations, how long?

Why 105 of cells were seeded in each well 96-well plate? What it comes from? Justify in the text – the justification that a laboratory usually does this is unscientific.

Paragraphs 4.4.-4.7.

There is no description of how the compounds were tested. In what concentrations, how long, how they were seeded ...

Why such concentrations were selected to these studies? On the basis of what. Give justification in the text.

Results

Figures 2-4 – give references in the caption.

Table 1: give statistics. What does it mean “> 200”? Give full names of tested compounds.

Table 1 is not according to Molecules demands.

Figure 4: The charts are crookedly fit with the rest.

The figures are out of focus, the bars could be coloured – they would be better visible.

Conclusions

In what doses the tested compounds are cytotoxic to tested cancer cells?

The authors should include changes in the text on all comments from the reviewer.

Author Response

Reviewer 3.

■ Still the descriptions especially in the methodology and results is very chaotic.

→ We revised and summarized the results and material method as below.

“2. Results

2.1 Anti-proliferative activity

Oregonin (1), hirsutenone (2), and hirsutanonol (3) were isolated from AS [20] (Fig. 1) and examined the anti-proliferative activity using two PCa cell lines (PC-3, LNCaP). Among the three compounds, MTT assay demonstrated that hirsutenone (2) had the most potent anti-proliferative activity (Table 1).

2.2. NF-κB inhibitory activity

 We evaluated the NF-κB inhibitory activity of the three diarylheptanoid isolated from AS on PC-3 and LNCaP cells. Western blotting demonstrated that the relative protein expression level of NF-κB was effectively reduced by treatment of both PCa cell lines with oregonin (1), hirsutenone (2) and hirsutanonol (3) (Fig. 2).

2.3. Apoptosis-inducing activity

We evaluated the apoptosis-inducing activity of the three compounds (1-3) at various concentrations against the two PCa cell lines using flow cytometry. And the oregonin (1) and hirsutenone (2) showed more potent apoptosis inducing activities than hirsu-tanonol (3) (Fig. 3).

2.4. DNA methylation activity

We examined DNA methylation activity in PCa cells using methylation-specific PCR. We only evaluated the DNA methylation activity of hirsutenone (2) which showed strongest activity among three compounds, demonstrating its enhancement of the unmethylated DNA content and potent reduction of the methylated DNA content in both PC-3 and LNCaP cells (Fig. 4).

  1. Materials and Methods

4.1. Plant material

Alnus sibirica Fisch. ex Turcz. (AS) leaves were collected from Mt. Cheonggye in Gwacheon, Gyeonggi province, Korea in June 1998 and its identity was confirmed by Prof. M.W. Lee (Pharmacognosy Lab, Laboratory of Pharmacognosy and Natural Product De-rived Medicine, College of Pharmacy, Chung-Ang University) [20].

4.2. General experimental procedure

Precoated silica gel 60 F254 plate (Merck, Darmstadt, Germany) was used thin layer chro-matography (TLC). The spots were detected by 10% H2SO4 by heating and spraying.

The column chromatography was conducted using Amberlite XAD-2 (20-50 mesh, Fluka), Sephadex LH-20 (75-230 μm, mesh, Pharmacia), MCI-gel CHP 20P (75-150 μm, Mitsubishi), ODS-A gel (230/70, 400/230, 500/400 mesh, YMCco).

IR spectrometer was used Shimazu IR-435 (Japan) and NMR spectrometer (Varian GEM-INI 2000(USA), 300MHz), 1H-(Brucker AMX-500 (Germany), 500MHz), EI-Mass spectromer (GC-MS/MS-DS, TSQ 700 (USA)), Negative FAB-Mass spectrometer (VG70-VSEQ (England), Polarimeter (Jasco DIP-370 (Japan)).

4.3. Phytochemicals

The air-dried stem barks of A. sibirica (5.6 kg) were extracted with 80% Me2CO at room temperature, 5 times. Concentration from removing the Me2CO under vacuum. After Me2CO evaporation, it was dissolved in H2O and filtered. The filtrate was enforced to an Amberlite XAD-2 with a H2O : MeOH (from 100:0 to 0:100), yielding 3 subfractions. In fraction of 2, using a Sephadex LH-20 column with a solvent gradient system of H2O : MeOH (from 100:0 to 0:100), yielded 3 subfractions (2-1, 2-2, 2-3). Fraction 2-2 was applied to column chromatography MCI gel CHP-20P column, and yielded compound (1) (3 g) and compound (2) (68mg). Fraction 3 was subjected to YMC ODS gel column with gradient system of H2O : MeOH (from 100:0 to 0:100), yielded compound (3) (60 mg).

4.4. Cell culture

The PCa cell lines PC-3 and LNCaP clone FGC; LNCaP.FGC were purchased from the Korean Cell Line Bank (KCLB, Seoul, Korea). And the PC-3 passage number was 28 and LNCaP was 24. These cells were grown at 37 °C in a humidified atmosphere (5% CO2) in Roswell Park Memo-rial Institute (RPMI) 1640 medium (Sigma-Aldrich, St. Louis, MO, USA) supplemented with 10% fetal bovine serum, 100 IU/mL penicillin G (Gibco BRL, Grand Island, NY, USA). The cells were used after being counted using, a hemocytometer.

4.5. Anti-proliferative activity

The PC-3 or LNCaP cells (approximately 105 cells/well) were seeded in 96-well plate and pre-incubated for 24 h and then treated with 20 μL of the samples (1000, 500, 250, 125, 62.5 μM of each sample) with serum free DMEM and incubated for a further 24 h at 37 °C. Next, the culture medium was replaced with 100 μL phosphate-buffered saline (PBS) containing 1 mg/mL of 3-(4,5-dimethylthiazol-2-yl)-2,5-diphenyltetrazolium-bromide (MTT), and the cells were incubated for a further 4 h. After the supernatant was removed, the MTT-formazan product was dissolved in 200 μL dimethyl sulfoxide, and the absorbance of the solution at 540 nm was measured using a microplate reader (TECAN, Salzburg, Austria) to determine the degree of cell proliferation.

4.6. Flow cytometry analysis of apoptosis

PC-3 and LNCaP cells (approximately 106 cells/well) were seeded in a 6-well plate and pre-incubated for 24 h and then treated with 200 μL of the samples (500, 250, 125 μM of each sample) with serum free DMEM and incubated for a further 24 h at 37 °C, and the cells were washed once in PBS. Next, the cells were resuspended in binding buffer (100 μL) and stained with fluorescein isothiocyanate-annexin V (5 μL) and propidium iodide (5 μL) (BD, Franklin Lanes, NJ, USA) for 15 min in the dark with vortex mixing. Fluorescence was analyzed using a fluorescence-activated cell sorter (BD), and the percentages of necrotic cells, early and late apoptotic cells, and viable cells were compared.

4.7. Protein extraction and western blotting

PC-3 and LNCaP cells (approximately 106 cells/well in 6-well plate) were pre-incubated for 24 h and then treated with 200 μL of the samples (500, 250, 125 μM of each sample) with serum free DMEM and incubated for a further 24 h at 37 °C, the cells were harvested and washed twice with PBS and lysed using 45 μL RIPA buffer (Thermo Fisher Scientific, Waltham, MA, USA) and 5 μL protease inhibitor cocktail (Thermo Fisher Scientific). The cell lysates were kept on ice for 5 min and centrifuged at 15,000 rpm for 15 min at 4°C. The concentrations of the extracted proteins in the supernatants were determined using a BCA Protein Assay Kit (Thermo Fisher Scientific). Equal concentrations of extracted proteins were resolved on 10% SDS-polyacrylamide gel and the separated proteins were transerred on to nitrocellulose membranes (Bio-Rad Laboratories), which were incubated in blocking buffer (Thermo Fisher Scientific) for 1 h to prevent non-specific binding. Subsequently, the membranes were incubated with primary antibodies (1:1000, Bio-Rad Laboratories and Sigma-Aldrich, St. Louis, MO, USA) for 2 h at room temperature (25°C), washed with Tris-buffered saline-Tween buffer, and incubated with horseradish peroxidase-conjugated secondary antibodies (1:1000, Santa Cruz Biotechnology, Santa Cruz, CA, USA) for 1 h at room temperature. After three washings, an enhanced chemiluminescence detection kit (Santa Cruz Biotechnology) was applied to expose the membranes to X-rays and obtain protein band images on a medical X-ray film (AGFA, Mortsel, Belgium). Band densities were analyzed using the ImageJ software (National Institutes of Health, Bethesda, MD, USA) to quantify relative protein expression.

4.8. DNA extraction and bisulfite conversion

PC-3 and LNCaP cells (approximately 106 cells/well) were seeded in 6-well plates and incubated for 24 h and then treated with 200 μL of the samples (1000, 500, 250 μM of each sample) with serum free DMEM and incubated for a further 24 h at 37 °C. After washing with PBS, the cells were centrifuged for 5 min, and DNA was extracted using the QIAamp DNA mini kit (QIAGEN, Hilden, Germany). The concentrations of extracted DNA were measured using a Nanodrop spectrophotometer, and 5 μg of DNA was subjected to bisulfite conversion using the Genetic MethylEasyXceed Rapid DNA Bisulphite Modification Kit (Genetic Signatures, NSW, Australia), according to the manufacturer’s instructions.”

■ There are many technical errors in the text (double spaces, lack of spaces etc.).

→ We checked and revised all the technical errors.

■ English language must be corrected by a native speaker.

→ We will receive an English editing from MDPI before publishing.

■ Introduction: Give research hypotheses – still not added.

→ We added hypotheses about the AS with prostate cancer as below.

We isolated several diarylheptanoids, which are characteristic components of the Alnus species [20]. And we screened 213 different kinds of local native plants for the ability to inhibit LNCaP, an androgen-sensitive prostate cancer cell line, and DU145, which is a more aggressive androgen-insensitive prostate cancer cell line [21]. And we reported cytotoxic activities of diarylheptanoids from Alnus species on various human and mouse cancer cell lines. [22]. And we also found that hirsutenone inhibits the activation of NF-κB involved in human mammary epithelial cells stimulated with TPA [23]. In this study, we evaluated inhibitory activities of proliferation, NF-κB, apoptosis and DNA methylation of three diarylheptanoids (1-3) isolated from AS on androgen-dependent (LNCaP) and androgen-independent (PC-3) prostate cancer cell lines through a series of experiments to determine their potential value of preventing PCa which may contribute to the chemopreventive effects exerted by these phytochemicals.

■ Materials and Methods

4.3. Cell culture

The description is very chaotic and disordered.

If both cell lines are grown similarly, there is no need to describe the culturing method separately, just mention the differences.

From the description now it is not clear how many cell lines were tested?

→ In material and Methods, we revised and summarized the cell culture as below. (4.3. Cell culture is change to 4.4. Cell culture)

1) 4.4. Cell culture After a comprehensive the information provided by the American type culture collection (ATCC) site and Korean cell line Bank (KCLB) about Prostate Cancer PC-3 and LNCAP, the growth environment of PC-3 and LNCAP cells was similar, so the experimental conditions of PC-3 and LNCAP were the same. When conducting the actual experiment, there was no change in cell shape and growth rate etc. even if incubation and seeding under the same conditions. There is only a difference in whether the two cell line (PC-3, LNCaP) depend on androgen, except that there is no difference when experimenting with culture, seeding, etc.

2) And we tested total two prostate cancer cell lines (PC-3 and LNCAP) and three repetitive experiments.

“4.4. Cell culture

The PCa cell lines PC-3 and LNCaP clone FGC; LNCaP.FGC were purchased from the Korean Cell Line Bank (KCLB, Seoul, Korea). And the PC-3 passage number was 28 and LNCaP was 24. These cells were grown at 37 °C in a humidified atmosphere (5% CO2) in Roswell Park Memo-rial Institute (RPMI) 1640 medium (Sigma-Aldrich, St. Louis, MO, USA) supplemented with 10% fetal bovine serum, 100 IU/mL penicillin G (Gibco BRL, Grand Island, NY, USA). The cells were used after being counted using, a hemocytometer.”

■ 4.4. MTT

The description is very chaotic and disordered.

Still there is no description of how the compounds were tested. In what concentrations, how long?

Why 105 of cells were seeded in each well 96-well plate? What it comes from? Justify in the text – the justification that a laboratory usually does this is unscientific.

Paragraphs 4.4.-4.7.

There is no description of how the compounds were tested. In what concentrations, how long, how they were seeded ...

Why such concentrations were selected to these studies? On the basis of what. Give justification in the text.

We revised and summarized 4.5.-4.8. as below. (the paragraphs 4.4.-4.7. are change to 4.5.-4.8.)

1) The concentration of samples (1000, 500, 250, 125, 62.5 μM of each sample) in MTT were treated and incubated for 24 h at 37 °C.

2) About the general information of the concentration, the sample was diluted 10 times with culture during the experiment. And we tested after a comprehensive information provided by the American type culture collection (ATCC) site and Korean cell line Bank (KCLB) about Prostate Cancer PC-3 and LNCAP, the growth environment of PC-3 and LNCAP cells. And the recommended seeding amounts for 96-wel well plate and 6-well plate that we used. In general, for 96-well plates, recommend 0.4–1*105/well and we considering factors such as the rate of growth of prostate cancer cells etc.. So it decided 105/well at 96-well plate and 106/well at 6-well plate. And compared to the Normal control group, these conditions completely valid.

3) About the experimental concentration, seeding and time, Anti-proliferative activity were tested with the PC-3 or LNCaP cells (approximately 105 cells/well) were seeded in 96-well plate and pre-incubated for 24 h and then treated with 20 μL of the concentration of samples (1000, 500, 250, 125, 62.5 μM of each sample) and incubated for 24 h at 37 °C.

4) Flow cytometry analysis of apoptosis and Protein extraction and western blotting were tested with the PC-3 or LNCaP cells (approximately 106 cells/well) were seeded in a 6-well plate and pre-incubated for 24 h and then treated with 200 μL of the samples (500, 250, 125 μM of each sample) and incubated for 24 h at 37 °C.

5) DNA extraction and bisulfite conversion were tested with the PC-3 or LNCaP cells (approximately 106 cells/well) were seeded in 6-well plates and incubated for 24 h and then treated with 200 μL of the concentration of samples (1000, 500, 250 μM of each sample) and incubated for 24 h at 37 °C.

→ About the reason for the concentration used in this experiment.

1) In Anti-proliferative activity, we conducted the preliminary experiments at 5 type concentrations (1000, 500, 250, 125, 62.5 μM) and the results showed concentration-dependent activity. And the final computational concentration of sample (100, 50, 25 μM) in the Table.1 was sufficiently effective and verified according to IC50.

2) DNA extraction and bisulfite conversion were conducted with concentrations (1000, 500, 250 μM) and the results also showed concentration-dependent activity.

3) As evidenced by the above experiments, flow cytometry analysis of apoptosis and protein extraction and western blotting experiments were tested with the concentration (500, 250, 125 μM).

“4.5. Anti-proliferative activity

The PC-3 or LNCaP cells (approximately 105 cells/well) were seeded in 96-well plate and pre-incubated for 24 h and then treated with 20 μL of the samples (1000, 500, 250, 125, 62.5 μM of each sample) with serum free DMEM and incubated for a further 24 h at 37 °C. Next, the culture medium was replaced with 100 μL phosphate-buffered saline (PBS) containing 1 mg/mL of 3-(4,5-dimethylthiazol-2-yl)-2,5-diphenyltetrazolium-bromide (MTT), and the cells were incubated for a further 4 h. After the supernatant was re-moved, the MTT-formazan product was dissolved in 200 μL dimethyl sulfoxide, and the absorbance of the solution at 540 nm was measured using a microplate reader (TECAN, Salzburg, Austria) to determine the degree of cell proliferation.

4.6. Flow cytometry analysis of apoptosis

PC-3 and LNCaP cells (approximately 106 cells/well) were seeded in a 6-well plate and pre-incubated for 24 h and then treated with 200 μL of the samples (500, 250, 125 μM of each sample) with serum free DMEM and incubated for a further 24 h at 37 °C, and the cells were washed once in PBS. Next, the cells were resuspended in binding buffer (100 μL) and stained with fluorescein isothiocyanate-annexin V (5 μL) and propidium iodide (5 μL) (BD, Franklin Lanes, NJ, USA) for 15 min in the dark with vortex mixing. Fluorescence was analyzed using a fluorescence-activated cell sorter (BD), and the per-centages of necrotic cells, early and late apoptotic cells, and viable cells were compared.

4.7. Protein extraction and western blotting

PC-3 and LNCaP cells (approximately 106 cells/well in 6-well plate) were pre-incubated for 24 h and then treated with 200 μL of the samples (500, 250, 125 μM of each sample) with serum free DMEM and incubated for a further 24 h at 37 °C, the cells were har-vested and washed twice with PBS and lysed using 45 μL RIPA buffer (Thermo Fisher Scientific, Waltham, MA, USA) and 5 μL protease inhibitor cocktail (Thermo Fisher Scientific). The cell lysates were kept on ice for 5 min and centrifuged at 15,000 rpm for 15 min at 4°C. The concentrations of the extracted proteins in the supernatants were determined using a BCA Protein Assay Kit (Thermo Fisher Scientific). Equal concentrations of extracted proteins were resolved on 10% SDS-polyacrylamide gel and the separated proteins were transferred on to nitrocellulose membranes (Bio-Rad Laboratories), which were incubated in blocking buffer (Thermo Fisher Scientific) for 1 h to prevent non-specific binding. Subsequently, the membranes were incubated with primary antibodies (1:1000, Bio-Rad Laboratories and Sigma-Aldrich, St. Louis, MO, USA) for 2 h at room temperature (25°C), washed with Tris-buffered saline-Tween buffer, and incubated with horseradish peroxidase-conjugated secondary antibodies (1:1000, Santa Cruz Biotechnology, Santa Cruz, CA, USA) for 1 h at room temperature. After three washings, an enhanced chemiluminescence detection kit (Santa Cruz Biotechnology) was applied to expose the membranes to X-rays and obtain protein band images on a medical X-ray film (AGFA, Mortsel, Belgium). Band densities were analyzed using the ImageJ software (National Institutes of Health, Bethesda, MD, USA) to quantify relative protein expression.

4.8. DNA extraction and bisulfite conversion

PC-3 and LNCaP cells (approximately 106 cells/well) were seeded in 6-well plates and incubated for 24 h and then treated with 200 μL of the samples (1000, 500, 250 μM of each sample) with serum free DMEM and incubated for a further 24 h at 37 °C. After washing with PBS, the cells were centrifuged for 5 min, and DNA was extracted using the QIAamp DNA mini kit (QIAGEN, Hilden, Germany). The concentrations of extracted DNA were measured using a Nanodrop spectrophotometer, and 5 μg of DNA was subjected to bisulfite conversion using the Genetic MethylEasyXceed Rapid DNA Bisulphite Modification Kit (Genetic Signatures, NSW, Australia), according to the manufacturer’s instructions.”

■ Figures 2-4 – give references in the caption.

We gave the references in Discussion part including Figure 2-4.

“3. Discussion

Observance of the 2018 World Cancer Research Fund/American Institute for Cancer Research (WCRC/AICR) cancer prevention recommendations and its relationship to PCa were evaluated and a total of 398 PCa cases and 302 controls were included [25].

In this study, we tested the PCa prevention effects including anti-proliferative activity, NF-κB inhibitory activity, apoptosis-inducing activity and DNA demethylation activity of the three diarylheptanoids isolated from AS in vitro.

Anti-proliferative activity of these three compounds were performed using two PCa cell lines, PC-3 and LNCaP. Among the three compounds, MTT assay demonstrated that hirsutenone (2) showed the potent anti-proliferative activity against both PC-3 and LNCaP cells [IC50, 65.53 on PC-3 and 109.14 on LNCaP]. (Table.1)

Nuclear factor kappa-light-chain enhancer of activated B cells (NF-κB) is a well-known negative regulatory factor of apoptosis, which controls DNA transcription, cytokine production, and cell survival. Dysregulation of NF-κB has been associated with inflammation, immune disease, and cancer [26-30]. In particular, the activation of NF-κB signaling is correlated with PCa progression in PCa cell lines and has been associated with more advanced stages, chemoresistance, and metastatic spread [31-34].

Our previous research reported that, hisutenone inhibits lipopolysaccharide-activated NF-κB-induced inflammatory mediator production. We investigated the effect of hirsutenone on lipopolysaccharide-induced inflammatory mediator production in keratinocytes in relation to the Toll-like receptor 4-mediated activation of the extracellular signal-regulated kinase (ERK) and NF-κB pathways [35].

The NF-κB inhibitory activity of the three diarylheptanoid (1-3) in PC-3 and LNCaP cells from Western blotting demonstrated that the relative protein expression level of NF-κB was more effectively reduced by treatment of both PCa cell lines with oregonin (1) and hirsutenone (2) than by treatment with hirsutanonol (3). (Fig. 2)

Apoptosis is a programmed form of cell death that differs from necrosis, which is a form of traumatic cell death induced by external factors, including disease [36-37]. Unlike necrosis, which may result in the uncontrolled release of inflammatory factors, apoptosis does not usually lead to inflammation [38]. Apoptosis plays a crucial role in tumor suppression and is considered to be the most useful target for cancer therapy [39-40].

The apoptosis-inducing activity at various concentrations towards the two PCa cell lines using flow cytometry, 1-3 showed apoptosis-inducing activity. The lower left part, represents viable cell, the lower right part indicate early apoptosis. The upper right part is late apoptosis, which indicates cells that have died due to apoptosis progressing and necrosis, which represents the upper left part, is a cell necrosis or pathologically dead part. 1-3 showed apoptotic activities. Especially at the concentration 50μM on PC-3, oregonin (1) and hirsutenone (2) have more potent apoptosis-inducing activities than hirsutanonol (3). (Fig. 3)

DNA methylation is a chemical process that results in the addition of methyl groups to a DNA molecule. In general, DNA methylation occurs only on the cytosines of CpG dinucleotides. In healthy somatic cells, approximately 80% of CpG dinucleotides are methylated, and are excluded from the defined regions [41-43]. However there are some CpG-rich sequences in somatic cells, called CpG islands, that are usually demethylated [44- 45]. In addition, methylated cytosine may convert to thymine by spontaneous deamination, resulting in a T/G permanent mismatch mutation due to complementary pairing of T with A. When DNA methylation occurs at promoter regions that are located near a gene transcription start site containing a CpG island, gene expression is inhibited, resulting in gene silencing [46-48]. Accumulating evidence has pointed to a potentially more important role of gene silencing than that of mutation in cancer [48]. Indeed, several genes have been identified to be silenced or activated in various cancer types, and a large proportion of gene silencing results from DNA methylation. Some methylated genes, such as GSTP1, APC, RASSF1, CD44, and CDH1, have been associated with PCa [49]. In particular, methylation of GSTP1, which encodes an enzyme that is considered to play a key role in cancer susceptibility [52], has been detected in various cancers but shows high specificity for PCa and has been suggested to serve as a clinical biomarker for PCa diagnosis [24, 53-58].

DNA methylation activity in PCa cells using methylation-specific PCR, hirsutenone (2) was selected to evaluate its effect because 2 showed strongest anti-proliferative, NF-κB inhibitory, and apoptosis-inducing activities. DNA methylation activity of hirsutenone (2), demonsted its potent concentration-dependent enhancement of the unmethylated DNA content and reduction of the methylated DNA content in both PC-3 and LNCaP cells. (Fig. 4)

Conclusively, we demonstrated that hirsutenone (2), one of the active ingredients among compounds of AS, is the most effective for prevention of cancer. These results suggested that 2 is promising agent to treat and prevent of PCa and in vivo experiment will be required.”

■ Table 1: give statistics. What does it mean “> 200”? Give full names of tested compounds.

→ We screened 213 plants [21] and 45 compounds (unpublished data) about anti-proliferative activity against prostate caner cell lines. And we selected Alnus sibirica and three diarylheptanoid (1-3) among them. The meaning of >200 is negative activity against the cancer cells. And we conducted 3 times test but neglected statistics in this case.

■ Table 1 is not according to Molecules demands.

We changed Table 1. according to Molecules. You can check in manuscript.

■ Figure 4: The charts are crookedly fit with the rest.

We fixed the crooked one and replaced Figure 4 with high resolution and color pictures. You can check in manuscript.

■ The figures are out of focus, the bars could be coloured – they would be better visible.

We replaced all the figure charts with high resolution and color pictures. You can check in manuscript.

■ Conclusions

In what doses the tested compounds are cytotoxic to tested cancer cells?

The doses of tested compounds were 20 μL with different concentrations. (Table. 1)

■ The authors should include changes in the text on all comments from the reviewer.

→ We answered all the comments raised by other reviewer.
